# Microspheres from light–a sustainable materials platform

**Laura Delafresnaye** [1,2,4] ✉, **Florian Feist** [3,4], **Jordan P. Hooker**[1,2] & **Christopher Barner-Kowollik** [1,2] ✉

Driven by the demand for highly specialized polymeric materials via milder, safer, and sustainable processes, we herein introduce a powerful, purely light driven platform for microsphere synthesis – including facile synthesis by sunlight. Our light-induced step-growth precipitation polymerization produces monodisperse particles (0.4–2.4 μm) at ambient temperature without any initiator, surfactant, additive or heating, constituting an unconventional approach compared to the classically thermally driven synthesis of particles. The microspheres are formed via the Diels-Alder cycloaddition of a photo-active monomer (2-methylisophthaldialdehyde, MIA) and a suitable electron deficient dienophile (bismaleimide). The particles are stable in the dry state as well as in solution and their surface can be further functionalized to produce fluorescent particles or alter their hydrophilicity. The simplicity and versatility of our approach introduces a fresh opportunity for particle synthesis, opening access to a yet unknown material class.

Polymeric particles are a versatile class of materials due to their small size (nanometer to micrometer), high volume-surface area ratio, and tunable properties, therefore finding applications in fields as diverse as electronics, catalysis, drug delivery, bio-sensors, coatings, bio-imaging, exfoliating/texturing agents in cosmetics, or even as standards to calibrate analytical devices[1–3]. Specifically, microspheres are spherical particles ranging from 0.1 to 100 μm in size, and they can be easily separated from many media by centrifugal density separation, filtration, or even via magnetic separation. Further, microspheres' surface can be readily functionalized with various substrates such as proteins, nucleic acids, and dyes, making them attractive for point-of-care diagnostic devices. As a prime example, microspheres or latex agglutination tests (LATs) have been developed in the late 1960's and employed in home pregnancy tests[4], the detection of myocardial infarction[5] and, more recently, for the rapid diagnosis of SARS-CoV-2 infections[6]. Microspheres can be employed either as a solid material or as a stable suspension in liquids and their size, morphologies, and their surface reactivities are dictated by their preparation method.

Classically, polymeric particles are produced by thermal processes in dispersed media requiring heating and cooling periods in addition to temperature regulation throughout the polymerization, therefore requiring substantial amounts of energy, often causing problems of local overheating and uncontrolled reaction. Photochemistry is a viable tool, offering more efficient, safer, and milder reactions conditions as it allows for temporal control of the polymerization by simply switching the radiation source on and off. By controlling the light intensity, wavelength, and reactor design, unique polymer architectures and properties have become accessible[7]. However, the implementation of photochemistry from solution to dispersed media to produce particles is far from trivial. The most prominent challenge is to overcome the turbidity of particles in dispersed media and the scattering/absorption of the photoinitiators and monomers. Indeed, the particle size is a key parameter which critically controls the light scattering and the radiation penetration depth throughout the sample, resulting in the potential loss of photons, incomplete conversions and lack of control[8]. Jasinski et al. recently collated a comprehensive overview of the state-of-the-art of

[1]School of Chemistry and Physics, Queensland University of Technology (QUT), 2 George St, Brisbane 4000 QLD, Australia. [2]Centre for Materials Science, Queensland University of Technology (QUT), 2 George St, Brisbane 4000 QLD, Australia. [3]Institute of Nanotechnology (INT), Karlsruhe Institute of Technology (KIT), Hermann-von-Helmholtz-Platz 1, 76344 Eggenstein-Leopoldshafen, Germany. [4]These authors contributed equally: Laura Delafresnaye, Florian Feist. ✉e-mail: laura.delafresnaye@qut.edu.au; christopher.barnerkowollik@qut.edu.au

photopolymerization in dispersed systems and covered the main homogeneous and heterogeneous processing techniques of this emerging technology[9]. Methods reported so far predominantly focus on photopolymerization employing photoinitiators such as type I and II radical photoinitiators, photoacid generators, and photo(redox) catalysts. Photoinitiated polymerization-induced self-assembly (photo-PISA)[10] and photoinduced electron transfer-reversible addition–fragmentation chain transfer polymerization (PET-RAFT)[11] are the most established processes and have generated considerable interest for their high degree of control over the macromolecular architecture, which can generate complex particle morphologies such as spheres, vesicles and worms under mild conditions (e.g. ambient temperature, visible light). Adaptations of thermally driven polymerization techniques to photopolymerization techniques have mostly been achieved employing emulsion, dispersion and mini-emulsion techniques. These techniques inherently require the use of process additives as well as relatively complex process control to produce polymer particles efficiently and effectively. Specifically, process additives such as stabilizers/emulsifiers and initiators will carry over to the final polymer product and often be present as an undesirable contaminant.

Rarely considered, precipitation polymerization is a subclass of homogeneous polymerization, where the initial mixture is homogeneous and turbidity only ensues during the polymerization. Critically, this method does not require any surfactants or additives and simply requires a monomer, a cross-linker and an initiator dispersed in a Θ-solvent (i.e., the solvent must dissolve all the reactants before the reaction but behave as a non-solvent for the polymer). The precipitation polymerization technique has been generally described by Stöver[12], yet only a handful of reports exploit photochemistry, exclusively relying on radical photoinitiators and chain growth polymerization[13,14]. However, free radical polymerization and the inherent generation of radicals cause low oxygen tolerance, thus leading to interference with sensitive species – for example a bioactive species. There is thus a growing demand for light-induced alternative polymerization techniques such as step-growth polymerization[15]. In that regard, light-mediated thiol–ene/yne polymerization has been successfully implemented to produce polymeric particles. For example, Jasinski et al. reported the preparation of poly(thioetherester) latex nanoparticles employing a dithiol/diene step-growth mechanism in miniemulsion[16]. Tan et al. also produced uniform particles in the 2–8 µm range via a thiol-isocyanate ligation reaction in

dispersion polymerization[17]. Recently, our team introduced a platform technology relying on a range of photoinduced crosslinking reactions of post-functionalized pre-polymers in a precipitation-like process. The concept utilizes low molecular weight pre-polymers (i.e. poly(methyl methacrylate), polystyrene) containing photoactive moieties along the chain which will cross-link and form particles upon light irradiation (λ = 300–525 nm). Stable monodisperse particles were obtained employing the Nitrile-Imine mediated Tetrazole-Ene Cycloaddition (NITEC)[18], the [4 + 2] and [4 + 4] cycloaddition of ortho-methylbenzaldehydes (oMBAs)[19], as well as triazolinedione (TAD) chemistry[20]. Interestingly, multi-functional particles with inherent / on-demand fluorescence, and on-demand degradation were readily prepared by varying the photoactive partner[21,22].

Herein, we initiate step-growth photopolymerization from monomeric building blocks by exploiting the photoinduced Diels-Alder (DA) cycloaddition of light-generated ortho-quinodimethanes (o-QDMs) with maleimides, thus introducing the simplest photochemical particle technology to date. The light-induced particle synthesis only relies on a 1:1 mixture of photoactive monomer (i.e., AA monomer) and a suitable partner (i.e., BB monomer) at ambient temperature without any additives, initiators, or surfactants (Fig. 1). Critically, we exploited the sun as light source, thus taking advantage of an unlimited and free resource. The enticing prospect of producing a key material class by clever use of a natural and unlimited resource can critically contribute to an advanced sustainable economy. Often regarded as harmful, our sun is a powerful and free resource. For example, our home country Australia, is one of the sunniest countries in the world with 1387 to 2264 kWh/m² a year, representing 3.8 to 6.3 peak sun hours a day (1 peak sun hour = 1000 W/m² of sunlight per hour).

## Results
### Monomer design and synthesis
Upon irradiation, 2-substituted benzaldehydes−conventionally termed photoenols−undergo a photoenolization to generate highly reactive o-QDMs which can subsequently react with a dienophile partner via [4 + 2] Diels-Alder cycloaddition. Due to their high reactivity, efficiency and versatility, photoenols have been successfully employed for laser lithography[23], sequence-defined copolymer[24] and particle synthesis[19]. Recently, our team synthesized methylisophthalaldehyde (MIA) derivatives, therefore generating two equivalents of o-QDMs from a single aromatic ring which were subsequently reacted with maleimide[25]. In depth investigation of the Diels-Alder reactions revealed that – for

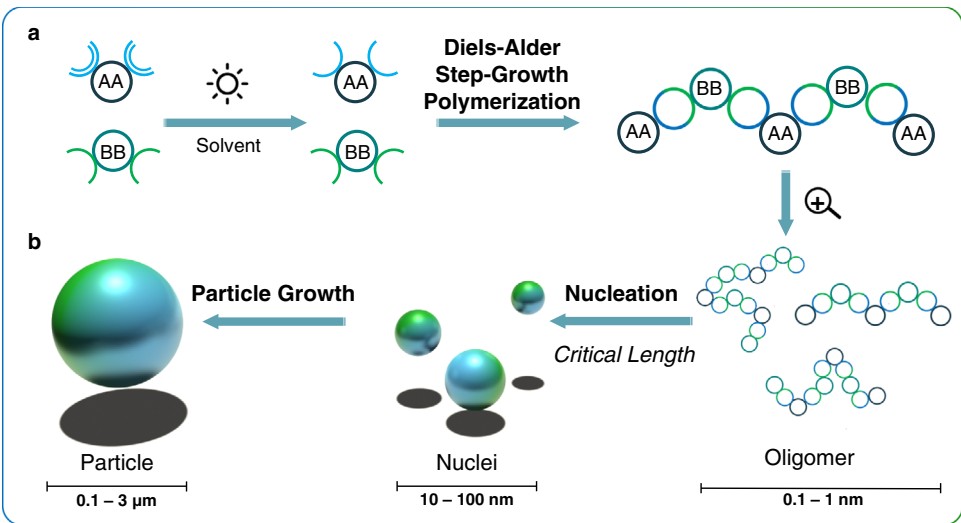

**Fig. 1 | Principle of light induced particle formation. a** Under irradiation, an AA monomer forms a reactive intermediate, which reacts with a BB monomer in a Diels-Alder step-growth polymerization. **b** As the linear polymer chains grow, they precipitate from solution to form particles.

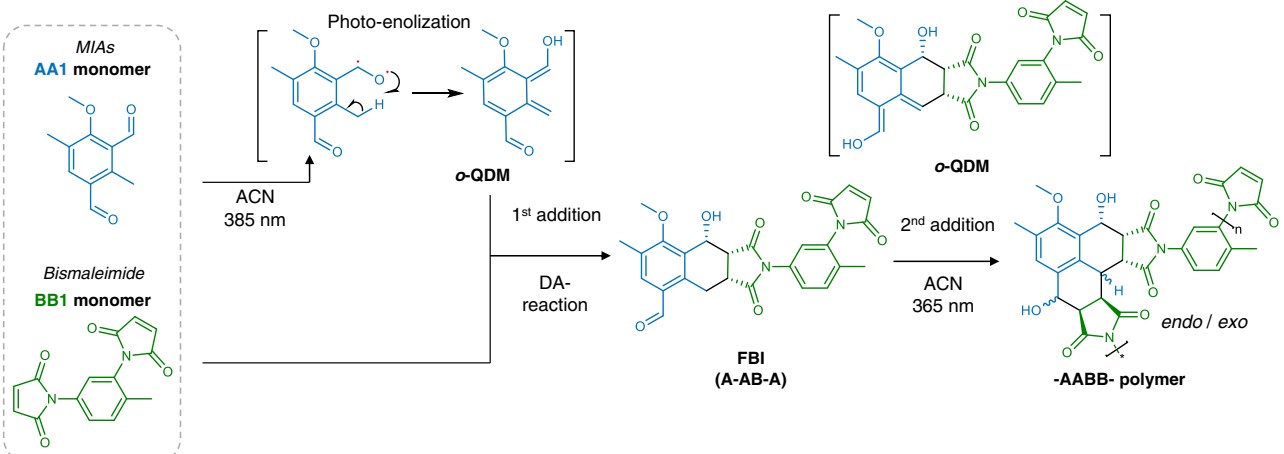

**Fig. 2 | Photopolymerization Reaction Mechanism.** Photoenolization of methy-lisophthalaldehyde (MIA) generates a reactive *ortho*-quinodimethane (*o*-QDM) which reacts with the maleimide moiety to form a benzo[*f*]isoindole-5-carbaldehyde (FBI) adduct. The second *o*-QDM subsequently reacts with a maleimide moiety to form a AABB polymer. Example employs **AA1** and **BB1** monomers.

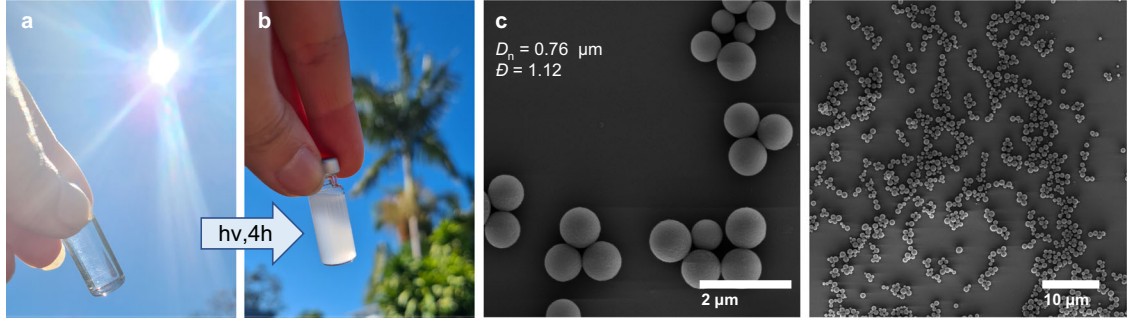

**Fig. 3 | Particle formation under sunlight. a** Reaction mixture before and (**b**) after 4 h of sunlight irradiation. **c** corresponding SEM images of the particles produced in Australian sunlight (run 1.1_C, Supplementary Table 1).

asymmetric MIAs – the first maleimide addition occurs at 385 nm at the *ortho*-formyl position while the second addition occurs at 365 nm at the *para*-formyl position once the first formyl group is converted. It is worth mentioning that the reaction predominantly leads to the *endo*-adducts especially in polar solvents such as acetonitrile; however, the diastereoselectivity is reduced by non-planarity and increasing stereochemical bulk of the N-substituent[26]. Herein, we employ the MIA as AA monomer and a bismaleimide as BB monomer to produce an –AABB– step growth polymer via a single irradiation step employing an LED covering the 365–385 nm range or direct sunlight (Fig. 2, Supplementary Fig. 2).

For a step-growth polymerization of an AA/BB system, the stoichiometry, selectivity and the reaction times need to be carefully chosen to obtain high molecular weight polymers. Firstly, 4-methoxy-2,5-dimethylisophthalaldehyde was selected as the photoactive monomer (**AA1**) and was synthesized from 2,5-dimethylphenol via a *Duff*-reaction and subsequent base mediated etherification. The rationale underpinning the design of monomer **AA1** entails several observations and findings from earlier studies. Even if the thioether photoenol's light absorption is shifted towards visible light[27,28], ether photoenols were preferred for their higher quantum yield, lifetime and reactivity. Commercially available 2,4-toluene bismaleimide was selected as the **BB1** counterpart.

## Photochemical polymerization and particle formation

In conventional precipitation polymerization, a monomer, a cross-linker and an initiator are dispersed in a Θ-solvent (typically acetonitrile), and the reaction starts as a homogeneous solution (stage 0). After initiation, the soluble oligomer chains grow via chain growth radical polymerization until they reach a critical length at which they are no longer soluble and separate from the continuous medium by entropic precipitation (the crosslinker prevents the polymer and solvent from freely mixing) to form nuclei (stage I)[29]. After the nucleation period, the particles continue to grow from its surface by capturing oligomers from solution to yield micrometer-sized polymer particles (stage II). Generally, precipitation polymerization is performed under gentle agitation to avoid coagulation, for instance with a shaking-bed or rotary evaporator. Herein, while the precipitation polymerization stages are similar, the underlying mechanism differs and relies on a step-growth polymerization. We simply dissolve 1 eq. of **AA1** and 1 eq. of **BB1** monomer in acetonitrile (ACN, 2.5 mmol L$^{-1}$) and place the vial on a bottle roller under direct sun irradiation (Supplementary Fig. 1). As displayed in Fig. 3a, the initial mixture is clear and homogenous (stage 0), yet becomes turbid after 4 h of sunlight irradiation (Fig. 3b). The particles are collected by centrifugation and washed several times with THF and ACN to remove any soluble oligomers and unreacted monomers (i.e. supernatant). The particles are narrow disperse with a number-average diameter ($D_n$) of 0.76 μm and a dispersity $Đ$ of 1.12 as shown in the Scanning Electron Microscopy (SEM) images (Fig. 3c). The same experiment was performed under a 3 W 365 nm LED in a controlled laboratory environment and yielded comparable results ($D_n = 0.79$ μm, $Đ = 1.15$) (run 1.1_A, Supplementary Table 1, Supplementary Fig. 3). A 10 W LED was also employed and the experiments produced larger particles ($D_n = 1.06$ μm) with a very low dispersity ($Đ = 1.03$, run

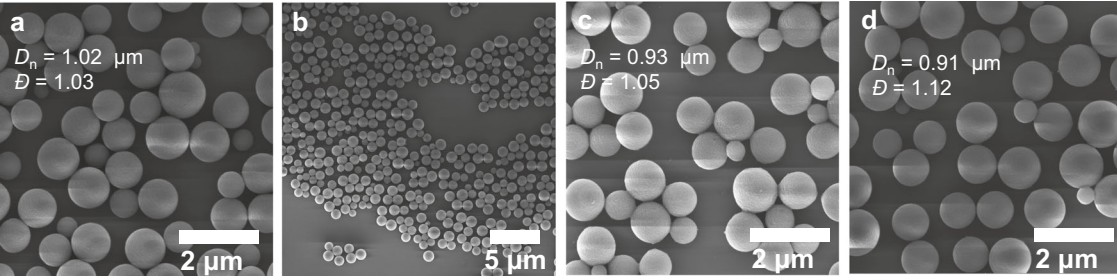

**Fig. 4 | Particle aging at ambient temperature.** SEM images and corresponding number-average diameter $D_n$ and dispersity $Đ$ of the **AA1/BB1** particles after 6 months at ambient temperature (**a**, **b**) in acetonitrile, **c** tetrahydrofuran, and **d** chloroform (run 1.1_B).

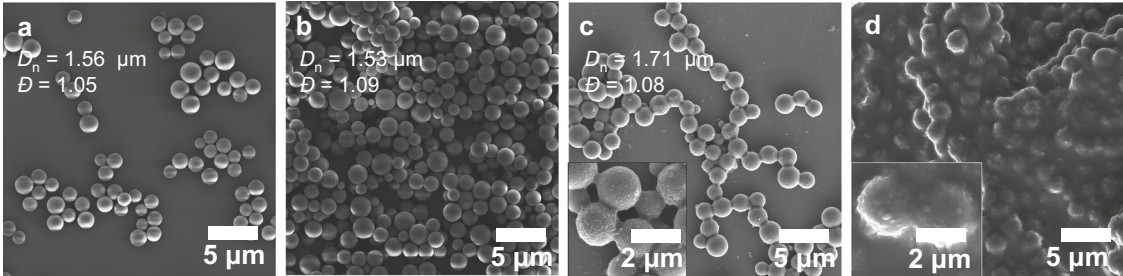

**Fig. 5 | Particle aging at elevated temperatures.** SEM images and corresponding number-average diameter $D_n$ and dispersity $Đ$ of the particles **AA1/BB1** (run 1.1_D, $D_n = 1.55\,\mu m$, $Đ = 1.08$). **a** Dry particles kept at 150 °C for 1 month and redispersed in acetonitrile. **b** After 1 month in trichlorobenzene (TCB) at 25 °C. **c** After 2 weeks and (**d**) 1 month in trichlorobenzene at 150 °C.

1.1_B). Our team recently developed a methodology to track the early stages of polymer particle nucleation and growth by employing the backscattering of a laser irradiation[30]. The real-time tracking of particle growth was compared to the final particles size measured by SEM and we observed that the LED power as well as the wavelength govern the final particle diameter. This trend could thus explain the differences observed in particle size when different light sources are employed, yet its full exploration is beyond the scope of the present study.

Since the final microspheres are not soluble in common solvents, we analyzed the residual supernatant via Size Exclusion Chromatography (SEC) and SEM. SEC reveals that the supernatant contains the **AA1** and **BB1** monomers as well as larger oligomers up to 6000 g mol⁻¹ (Supplementary Fig. 4A). We can thus assume that the −AABB− polymer reaches its critical length when it precipitates from solution at 6000 g mol⁻¹—a rather low molecular weight for a polymer. The steric bulk of the repeating unit in the −AABB− polymer, the binding angle of the bismaleimides, the hydrogen-donor and acceptor properties and the dipole moment contribute to the rapidly decreasing solubility and high tendency to form stable latices. SEM images of the supernatant show residual polymers and, occasionally, very small particles that were too small to be collected by centrifugation (Supplementary Fig. 5).

**Particle stability**
Remarkably, the particles are formed without any crosslinker and we thus investigated their stability in various solvents and conditions. Initially, we redispersed the particles in ACN, THF, and chloroform and stored them at ambient temperature (25 °C) for up to 6 months (refer to pictures in Supplementary Fig. 8). Importantly, the particle size in these 3 solvents lies between 0.96 and 1.06 μm ($Đ = 1.03–1.08$), thus the particles do not shrink or swell in polar/non-polar solvents. SEM analyses revealed that the particles retained their integrity and sphericity over the 6 months period in the 3 tested solvents ($D_n = 0.91$, 0.93, 1.02 μm) whilst retaining a low dispersity (Fig. 4, Supplementary Figs. 6 and 7, Supplementary Table 2).

We subsequently exposed the particles to harsher conditions. Since the particles are collected as a dry powder, we stored them at 150 °C for 1 month under ambient atmosphere and redispersed them in ACN for SEM analyses (Fig. 5a). Compared to time 0 (run 1.1_D, $D_n = 1.50\,\mu m$, $Đ = 1.07$), the particles show no sign of degradation and remained stable even at high temperatures ($D_n = 1.56\,\mu m$, $Đ = 1.05$). As mentioned, polymeric microspheres are extensively used as chromatographic material in high-performance liquid chromatography and other interaction-based chromatography such as SEC. High temperature SEC usually employs 1,2,4-trichlorobenzene (TCB) as the mobile phase at 140 °C to characterize polyethylene materials for instance. Therefore, we were interested to push the boundary of the microspheres' stability and redispersed them in TCB. We stored them at ambient temperature and at 150 °C for 1 month (Fig. 5b–d, Supplementary Fig. 9). The microspheres initially measured 1.55 μm ($Đ = 1.08$) at time 0 and 1.53 μm ($Đ = 1.09$) after 1 month in TCB at ambient temperature, demonstrating that (*i*) the particles are stable in TCB for at least 1 month, and (*ii*) the particles do not shrink/swell if dispersed in TCB compared to ACN in the previous experiment (run 1.1_D). When heated at 150 °C in a harsh solvent (i.e. TCB), the surface of the particles became coarse, yet the microspheres remained spherical and individuals. However, after 1 month, significant coalescence occurred and the particles were no longer stable. Particle counting revealed that some coalescence occurred earlier since, after 1 week and 2 weeks, the particles' size increased from 1.55 μm to 1.65 and 1.71 μm, respectively (Supplementary Table 2).

In addition, Differential Scanning Calorimetry (DSC) analyses did not show any crystallinity or phase transition before decomposition (Supplementary Fig. 10). Thermogravimetric analyses (TGA) were conducted up to 800 °C: a 2% weight loss appears between 25 and 210 °C and is attributed to trapped water which can be formed by E1 elimination of the OH groups (Supplementary Fig. 11). Then, a steady decomposition begins at 360 °C with a final weight loss of 47.7% (residual carbon content is about 54%).

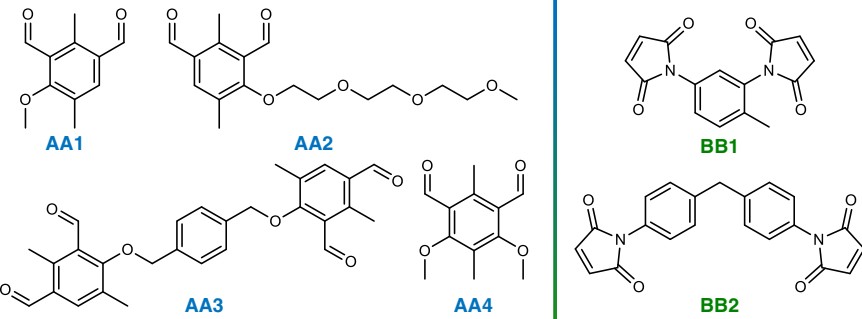

**Fig. 6 | Particle surface functionalization. a** NITEC reaction with the residual maleimide moieties of the **AA1/BB1** particles (run 1.1_E) and 4-(2-phenyl-2H-tetrazol-5-yl)benzoic acid (Tz1) or 4-(2-(4-methoxyphenyl)−2H-tetrazol-5-yl)benzoic acid (Tz2). Pictures of the isolated fluorescent particles in dry state and redispersed in THF. **b** Thiol-ene reaction with the residual maleimide moieties of the **AA1/BB1** particles (run 1.1_F) and PEG-SH. Pictures of the unfunctionalized particles (left) and PEG-particles (right) in water.

**Fig. 7 | Various AA and BB Monomers. AA1**−4-methoxy-2,5-dimethylisophthalaldehyde, **AA2**−4-(2-(2-(2-methoxyethoxy)ethoxy)ethoxy)-2,5-dimethylisophthalaldehyde, **AA3**−4,4′-((1,4-phenylenebis(methylene))bis(oxy)) bis(2,5-dimethylisophthalaldehyde), **AA4**−4,6-dimethoxy-2,5-dimethylisophthalaldehyde, **BB1**−2,4-toluene bismaleimide, **BB2**−1,1′-(methylenedi-4,1-phenylene)bismaleimide.

## Particle surface functionalization

Critically, we investigated the potential applications of the particles by exploring their residual functional surface groups. Since there are no additives or surfactants present, the particles' surface is free from contaminants and is readily available for surface chemistry. Even at 1:1 ratio, residual monofunctional maleimide remains and can be exploited for further reactions. Indeed, maleimides show high activity in cycloadditions and are very often employed as electron-poor dienophiles. For example, fluorescent pyrazoline adducts can be formed employing tetrazoles via the Nitrile-Imine mediated Tetrazole-Ene Cycloaddition (NITEC) reaction. Tetrazoles have already been employed to ligate a wide variety of substrates in biological and materials contexts, and their versatility was shown by application in fields as diverse as single-chain nanoparticles[31], direct laser writing[32], or self-healing hydrogels[33]. We thus selected two tetrazoles—one bearing a phenyl end group (Tz1) and one bearing a methoxy phenyl (Tz2)—and employed UV-B light to trigger the photoinduced cycloadditions with the residual surface maleimide groups (Fig. 6a). Employing mild conditions (i.e. ambient temperature, UV-light) and in less than 30 minutes, the reaction mixtures exhibited the characteristic fluorescence of the pyrazoline adducts—i.e., blue for Tz1 and yellow for Tz2. After several washes, the particles were isolated by simple centrifugation and exhibited inherent fluorescence at $\lambda_{max} = 480$ nm for Tz1 and 530 nm for Tz2 as shown in dry state and dispersed in THF (Fig. 6a, Supplementary Figs. 12 and 13). SEM analyses showed that the particles

are stable with no changes in the particles size and dispersity (Supplementary Fig. 12, Supplementary Table 3). Importantly, the $R_2$ group of the tetrazoles remains intact and could serve for further post-functionalization: herein we employed tetrazoles bearing an acid group, yet alcohol and acrylate tetrazoles as well as more specialised polymers, peptides and even protein-functionalised variants are also readily available[34,35]. As a second approach, we exploited the residual maleimide for a thiol-ene reaction with a poly(ethylene glycol) methyl ether thiol (PEG-SH, 2000 g mol⁻¹). The particles were simply mixed with a PEG-SH at ambient temperature and recovered by centrifugation. We were thus able to easily disperse the isolated particles in water, while the particles aggregated and settled before the functionalization (Fig. 6b). It is worth noting that SEM analyses showed that the particle size slightly increased by about 100 nm (Supplementary Table 3). Since many peptides and drugs contains thiols – which would react in the presence of radicals – this model experiment with a PEG thiol highlights the versatility of these particles for tailor-made applications.

## Polymer backbone alteration

Another approach to incorporate functionality without further functionalization is to alter the polymer backbone by judiciously varying the AA and BB monomers (Fig. 7). We incorporated a PEG side-chain to further increase the solubility of the AA monomer by synthesizing 4-(2-(2-(2-methoxyethoxy)ethoxy)

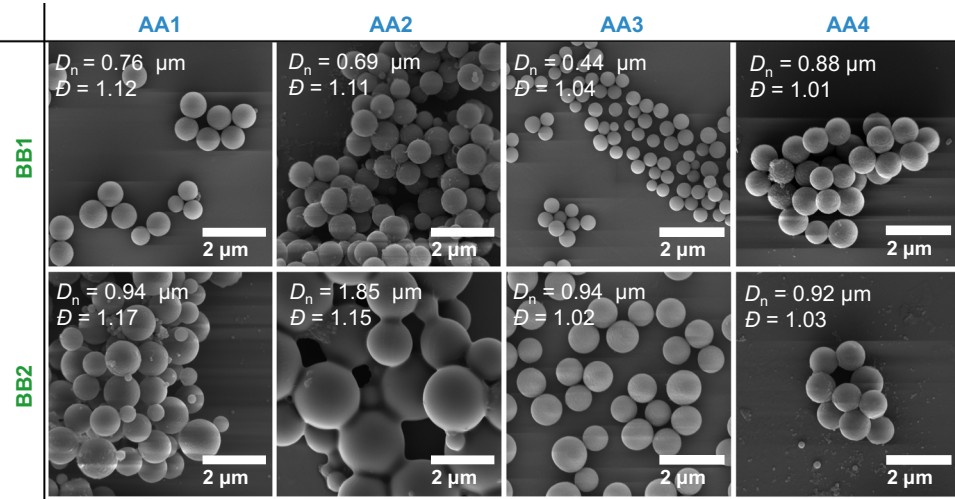

**Fig. 8 | SEM images of various AA and BB particles.** SEM images and corresponding number-average diameter $D_n$ and dispersity $Đ$ of particles produced under sunlight ([AA] = [BB] = 2.5 mmol L⁻¹, ACN, Supplementary Table 1).

ethoxy)-2,5-dimethylisophthalaldehyde (**AA2**). A cross-linker monomer generating four $o$-QDMs was also synthesized: 4,4'-((1,4-phenylenebis(methylene))-bis(oxy))bis(2,5-dimethylisophthalalde-hyde) (**AA3**). Initially, non-symmetric AA monomers were syntheti-cally more feasible. In the meantime, the synthesis of symmetric AA monomers could be improved, and we herein show their usefulness in the particle synthesis and selected 4,6-dimethoxy-2,5-dimethyli-sophthalaldehyde as monomer **AA4**. We also varied the BB mono-mer by employing another commercially available linker 1,1'-(Methylenedi-4,1-phenylene)bismaleimide (**BB2**).

The different AA/BB monomer combinations were irradiated under sunlight, producing stable microspheres. Interestingly, we noted a significant change in the particles' size as reported in Supplementary Table 1 and Fig. 8. As mentioned, **AA2** is the most soluble AA monomer and the molecular weight at which the oligomer precipitates is close to 10,000 g mol⁻¹ (run 2.2, Supplementary Fig. 4)–whereas the oligomers critical molecular weights are close to 6000 g mol⁻¹ for the other AA monomer–ultimately leading to larger particles ($D_n = 1.85$ μm). **AA3** produces smaller particles (run 2.1, $D_n = 0.44$ μm) and the SEC reveals that the supernatant does not contain oligomers above 2000 g mol⁻¹, therefore confirming the correlation between the oligomers' critical molecular weight and the particle size. **AA4** has a structure and solu-bility similar to **AA1** and produces particle in the same size range (run 4.1, 4.2). Another trend shows that the **BB2** monomer leads to bigger particles than **BB1** monomer and broader dispersity. This could be explained by the structure of the **BB2** monomer in which the mal-eimides are far apart, resulting in an **AABB2** backbone likely less compact than the **AABB1**. Since the polymeric backbone will alter the particles' surface reactivity, their solubility and stability, it would be interesting to perform in-depth investigations of the influence of the backbone on the particle size by varying the AA/BB concentrations and ratio, solvent combinations and light intensity, yet this beyond the scope of the present study. By adjusting the polymer backbone, we envision to tailor the particles' inherent properties such as degrad-ability, chemiluminescence, or even conductivity. We also anticipate that more detailed studies would readily facilitate the optimisation of particle size and dispersity for targeted systems and applications.

## Summary & conclusion
Photopolymerization in dispersed media is a hybrid technology that combines the ecological and efficiency advantages of two critical fields. Herein, we introduce a simple and purely light driven platform for microsphere synthesis. Relying on the Diels-Alder cycloaddition

of light-generated *ortho*-quinodimethanes with a dienophile partner, we translated this photochemical reaction into a step-growth poly-merization in combination with a precipitation process to ultimately produce polymeric particles. Our method does not require any initia-tor, surfactants, additives, or heating, but only an equimolar ratio of two monomers dissolved in ACN. Within a few hours under sunlight, microspheres are formed by cross-linking the photoactive AA mono-mer and a suitable BB monomer partner. By adjusting the AA/BB pair, particles from 440 nm to 2.4 μm can be produced under mild condi-tions. Importantly, the particles' surface is free from contaminants and can be post-functionalized to readily generate fluorescent particles or modify their dispersibility, for instance in aqueous media. Critically, the particles are stable in various solvents such as ACN, THF, Chloro-form, and TCB at ambient temperature and withstand elevated tem-perature (150 °C) for extended time in their dry solid state. Our platform is unique in its significant dissimilarity from conventional polymer particle synthesis strategies. We argue that such systems will create new possibilities as advanced applications increasingly demand specialised materials – such as in biological applications or in-situ processes requiring simplicity, control and minimal processing. Importantly, our approach opens an avenue for generating a key material using a natural power source – our sun.

## Methods
### Synthetic procedures
Detailed synthetic procedures are described in the Supplementary Information and are accompanied with reaction schemes, SEM images, SEC spectra, DSC spectrum, TGA spectrum, description of the proto-cols and analytical methods, LC-MS and NMR characterizations figures.

### Particle Synthesis
For a typical reaction, stock solutions of AA and BB monomers were prepared in acetonitrile at a concentration of 5 mmol L⁻¹. Then, 1 mL of each solution was passed through a 2.5 μm PTFE syringe filter and placed in a crimp cap vial ($c_{AAmonomer}$ = $c_{BBmonomer}$ = 2.5 mmol L⁻¹; V = 2 mL). Oxygen was removed by passing through a stream of nitrogen (N₂) for 5 min. Under irradiation with a 3 W or 10 W LED (λ = 360–390 nm, 4 cm distance, Supplementary Fig. 2) on a Thermo-Fisher Scientific Bottle/Tube Roller at 10 rpm, the clear solution gra-dually becomes heterogeneous (Supplementary Fig. 1A). After 4 h, the turbid solution was centrifuged (15,000 rpm, 5 min), the supernatant was removed, and the solid pellet washed with THF twice. The result-ing particles were redispersed in ACN and characterized via SEM. For

larger scale, 10 mL of each solution was placed in a 20 mL crimp vial and the particles were centrifuged at 5,000 rpm for 5 min (Yield = 68.9%). For the sunlight experiments, the same conditions were used without the LED and by placing the bottle roller outside for 4–8 h (Supplementary Fig. 1B).

NB: **AA3** was used as 1.25 mmol L$^{-1}$ and BB as 2.5 mmol L$^{-1}$ to have 0.5 eq. (4 reactive moieties) for 1.0 eq. of bismaleimide (2 reactive moieties).

## Data availability

The authors declare that the data generated in this study are provided in the Supplementary Information and that raw data have been deposited in the KIT repository database (RADAR4KIT) under accession code (https://doi.org/10.35097/695).

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

## Acknowledgements

C.B.-K. acknowledges the Australian Research Council (ARC) for funding in the context of a Laureate Fellowship enabling his photochemical research program, as well as continued key support from the Queensland University of Technology (QUT) through the Center for Materials Science. C.B.-K. additionally acknowledges additional funding by the Deutsche Forschungsgemeinschaft (DFG, German Research Foundation) under Germany's Excellence Strategy for the Excellence Cluster '3D Matter Made to Order' (EXC–2082/1 – 390761711). The scanning electron microscopy analyses were enabled by use of the Central Analytical Research Facility (CARF) at QUT.

## Author contributions

All authors contributed to the discussion and evaluation of the results at all stages. J.P.H. and F.F. performed preliminary experiments. F.F. and L.D. conceived, designed, and performed the experiments. C.B.-K. motivated and supervised the study and was responsible for funding acquisition via an ARC Laureate Fellowship. L.D. wrote the manuscript draft, which F.F. and C.B.-K. subsequently edited and refined.

## Competing interests

The authors declare no competing interests.
