## [Peer Review File · Nature Communications]

Reviewers' Comments:

Reviewer #1:

Remarks to the Author:

The manuscript "Microspheres from Light – A Sustainable Materials Platform" by Barner-Kowollik and coworkers deals with the formation of microsphere particles employing a light-initiated ligation reaction. It is well written and presents a timely topic. Nevertheless, I have several comments that should be addressed in a revised version.

Comments:

1. The manuscript mainly deals with the synthesis of microspheres, which is of course a relevant topic. Nevertheless, I miss a critical discussion on the impact of the work.
2. How dispersible are these particles and how stable are dispersions?
3. The authors use remaining maleimide functions to functionalize the particle surface. Did the authors think about addition of functional groups to increase dispersibility or to introduce other properties? I would expect a bit more on this topic for a publication in Nat. Commun.
4. It seems like the particles/polymers are not soluble. Which solvents were tested to solubilize the particles?
5. Do the authors have an idea about polymerization conversion?
6. The authors discuss thermal stability of the particles. I would suggest to perform DSC and maybe TGA to support the statements.
7. Did the authors think about probing the porosity of the particles?
8. In Figure 1, nucleation and particle growth are illustrated, yet nucleation is not described in the manuscript. I suggest to follow the particle growth in time. It might be even useful to see whether nanoparticle are accessible as well.
9. In Figure S11 CDCl₃ should be CHCl₃, on page 2 it should be the CDCl₃ triplet.
10. page 12 in the SI "150C"
11. superscripts in the SI should be checked

Reviewer #2:

Remarks to the Author:

Barner-Kowollik et al have produced new polymeric particles by a light induced precipitation polymerisation. Here, the simple light driven Diels-Alder coupling of bismaleimide with photoactivated dialdehydes yields a polymer that precipitates to form polymeric particles. This step-growth polymerisation can be undertaken under mild conditions and produces monodisperse particles due to the mechanism of formation. However, the targeted application of these particles is still unclear. The particles have maleimide functionality on the surface (although this isn't quantified) which enables a range of different species to be attached to the surface. The authors have chosen to do a NITEC reaction to create fluorescent particles. This, however, does not require the mild conditions and surfactant free environment that makes these particles unique. Although I believe the new method and particles synthesised are of interest, I think the application currently lets them down. Further to

this I have a few questions regarding the rest modification and stability tests.

- Can the authors comment on the colloidal stability of the particles after modification. E.g. SEM/DLS before and after?
- Do the authors have an idea of how much of surface they can functionalise using the NITEC reaction. 50% of the surface should be maleimide therefore I would expect a much higher fluorescence intensity to be achieved.
- Is the stability test of the particles conducted with the modified particles?
 - If so, then please make this clearer.
 - If the stability tests are conducted before modification, I would suggest that the manuscript is reordered to put the functionalisation at the end. This would make the paper easier to read and follow.

Reviewer #1

The manuscript "Microspheres from Light – A Sustainable Materials Platform" by Barner-Kowollik and coworkers deals with the formation of microsphere particles employing a light-initiated ligation reaction. It is well written and presents a timely topic. Nevertheless, I have several comments that should be addressed in a revised version.

Comment 1: The manuscript mainly deals with the synthesis of microspheres, which is of course a relevant topic. Nevertheless, I miss a critical discussion on the impact of the work.

Reply: We thank the reviewer for their comment. In response, we added several sections to highlight the impact and versatility of our new platform technology (refer to the highlighted section in the manuscript).

Comment 2: How dispersible are these particles and how stable are dispersions?

Reply: We thank the reviewer for raising a critical aspect of our study. We have already conducted an extensive study on the stability of the dispersion in various solvents and presented the results in the manuscript. We have now added pictures of the dispersions at various interval up to 6 months in the Supporting Information section (refer to Figure S8). We further conducted an additional experiment in which we increase the particles' dispersibility in aqueous solvent by surface modification (also refer to comment #3).

Comment 3: The authors use remaining maleimide functions to functionalize the particle surface. Did the authors think about addition of functional groups to increase dispersibility or to introduce other properties? I would expect a bit more on this topic for a publication in Nat. Commun.

Reply: We thank the reviewer for their comment. We now carried out an additional experiment exploiting the residual maleimide moieties on the particles' surface to perform a thiol-ene reaction with a poly(ethylene glycol) methyl ether thiol (2000 g mol⁻¹). After their initial polymerization, the particles are barely dispersible in water and will immediately settle. However, after functionalization with the PEG chain, the particles are readily dispersible in water. We thus amended Figure 4 (now Figure 6) and added a section in the Supporting Information section (section 7.2, Figure S14, Table S3).

Furthermore, the last section of the study focuses on altering the particles' backbone by employing different monomers, thus showing another approach to introduce properties (e.g. degradability, solubility).

Comment 4: It seems like the particles/polymers are not soluble. Which solvents were tested to solubilize the particles?

Reply: As noted by the reviewer, the particles are not soluble. We tested many solvents (ACN, THF, chloroform, DMSO, DMF, DMAc, TCB, xylene) and only TCB at 150 °C was able to degrade the particles after a few weeks.

Comment 5: Do the authors have an idea about polymerization conversion?

Reply: We calculated a yield, which represents the conversion of monomer to particle, not a polymerization conversion. The yield is around 70% and we added a paragraph to the Supporting Information section (page 7) and to the main manuscript to note the yield. We expect that with suitably optimised reaction conditions such as light intensity/wavelength, reaction time, solvent and monomer systems, there is most certainly scope to optimise yields further.

Comment 6: The authors discuss thermal stability of the particles. I would suggest to perform DSC and maybe TGA to support the statements.

Reply: We thank the reviewer for their suggestion. We performed DSC and TGA analyses (Figures S10-11) and discussed the results.

Comment 7: Did the authors think about probing the porosity of the particles?

Reply: We thought about probing the porosity by Brunauer–Emmett–Teller analysis or by performing a microtome of the particles via Transmission Electron Microscopy, yet we have not conducted these analyses. Porogenic solvents are sometimes utilised in precipitation polymerization and controlling/inducing the porosity of the particles could be interesting, yet is out of the scope of the present study.

Comment 8: In Figure 1, nucleation and particle growth are illustrated, yet nucleation is not described in the manuscript. I suggest to follow the particle growth in time. It might be even useful to see whether nanoparticle are accessible as well.

Reply: We acknowledge the reviewer's suggestion which is based on a previous study we conducted (*ACS Macro Lett.*, **2021**, *10*, 851-856), where we followed the particle growth over time. As mentioned by the reviewer, it would be interesting to monitor the particle growth to control the size of the final particles with the light intensity, the reaction conditions and monomers' structure. However, such a study requires comprehensive experiments with a laser to screen all different parameters and is out of scope of the present study, where we focused on using sunlight. We will certainly follow-up with a detailed study on particle growth kinetics in a more specialized journal.

Comment 9: In Figure S11 CDCl₃ should be CHCl₃, on page 2 it should be the CDCl₃ triplet.

Reply: We thank the reviewer for pointing out this typographical error. We amended the Supporting Information section accordingly.

Comment 10: page 12 in the SI "150C"

Reply: We thank the reviewer for pointing out this typographical error. We changed "150C" to "150 °C" on page 12.

Comment 11: superscripts in the SI should be checked

Reply: We thank the reviewer for their suggestions and we have carefully checked the superscripts and subscripts.

Reviewer #2

Barner-Kowollik et al have produced new polymeric particles by a light induced precipitation polymerisation. Here, the simple light driven Diels-Alder coupling of bismaleimide with photoactivated dialdehydes yields a polymer that precipitates to form polymeric particles. This step-growth polymerisation can be undertaken under mild conditions and produces monodisperse particles due to the mechanism of formation. However, the targeted application of these particles is still unclear. The particles have maleimide functionality on the surface (although this isn't quantified) which enables a range of different species to be attached to the surface. The authors have chosen to do a NITEC reaction to create fluorescent particles. This, however, does not require the mild conditions and surfactant free environment that makes these particles unique. Although I believe the new method and particles synthesised are of interest, I think the application currently lets them down. Further to this I have a few questions regarding the rest modification and stability tests.

We thank the reviewer for their comments and we have amended the manuscript to highlight the versatility of our new method. Furthermore, we performed a thiol-ene reaction with the residual maleimide and a poly(ethylene glycol) methyl ether thiol (2000 g mol⁻¹) to allow the dispersibility of the particles in water. Both the NITEC and thiol-ene reactions were carried out at ambient temperature without any surfactant and the particles are recovered by simple centrifugation. We believe that these post-functionalization approaches are mild and open several avenues to tailor our particles to specific applications, e.g., in point-of-care applications or as chromatography materials.

Comment 1: Can the authors comment on the colloidal stability of the particles after modification.

E.g. SEM/DLS before and after?

Reply: We thank the reviewer for raising that point. The particles are stable after modification with the tetrazole and PEG-SH (added new experiment to demonstrate the impact of pegylation) and we have added pictures for visual inspection (Figure 6 and Figure S12). We further added SEM pictures before and after the modification step as well as the corresponding particles sizes in Figure S14 and Table S3.

Comment 2: Do the authors have an idea of how much of surface they can functionalise using the NITEC reaction. 50% of the surface should be maleimide therefore I would expect a much higher fluorescence intensity to be achieved.

Reply: We acknowledge the reviewer's comment, yet it is difficult to quantify the percentage of residual maleimide. As per step-growth polymerization's mechanism, the 1:1 stoichiometry of aldehyde and maleimide functional groups is not retained on the surface: the bismaleimide will react to form the polymer chain and the residual maleimide moieties on the surface are only associated with unreacted polymer chain ends. As the proportion of end chains is very low, surface-sensitive methods such as X-ray Photoelectron Spectroscopy or Time-of-Flight Secondary Ion Mass Spectrometry are analytically challenging in this case, and we therefore opted to qualify the presence of maleimide by chemical transformation.

Comment 3: Is the stability test of the particles conducted with the modified particles?

o If so, then please make this clearer.

o If the stability tests are conducted before modification, I would suggest that the manuscript is reordered to put the functionalisation at the end. This would make the paper easier to read and follow.

Reply: We thank the reviewer for raising that concern. The stability tests were conducted on the non-modified particles. We have thus re-ordered the manuscript as suggested with the stability presented first and the functionalization next. We also have re-numbered the figures accordingly.

Reviewers' Comments:

Reviewer #1:

Remarks to the Author:

The revised version of the manuscript addresses all my previous comments.

In particular, the comments regarding colloidal stability were addressed (reviewer 2 and mine) with additional measurements and the manuscript reordered to clarify the work.

Furthermore, the comments about surface functionalization (reviewer 2 and mine) were addressed. The authors made a fair point that physical methods are challenging to perform in order to achieve a quantification of reactive surface functionalities. Nevertheless, surface functionalization was assessed via chemical ligation. Additional experiments were added to the work to prove the surface functionality via fluorescence and dispersibility.

In my opinion the manuscript can be published.

Reviewer #2:

Remarks to the Author:

The authors have made minimal attempts to address the key concerns of the reviewers and hence the paper is more suitable to a more specialised journal such as macromolecules.